# Influence of Lisdexamfetamine Dimesylate on Early Ejaculation—Results from a Double-Blind Randomized Clinical Trial

**DOI:** 10.3390/healthcare9070859

**Published:** 2021-07-07

**Authors:** Mohammad Haghighi, Mona Doostizadeh, Leila Jahangard, Alireza Soltanian, Mohammad Faryadres, Kenneth M. Dürsteler, Annette Beatrix Brühl, Dena Sadeghi-Bahmani, Serge Brand

**Affiliations:** 1Behavioral Disorders and Substance Abuse Research Center, Hamadan University of Medical Sciences, Hamadan 6516848741, Iran; Haghighi@umsha.ac.ir (M.H.); mo.doostizadeh@gmail.com (M.D.); jahangard@umsha.ac.ir (L.J.); 2Modeling of Non-Communicable Diseases Research Center, Hamadan University of Medical Sciences, Hamadan 6516848741, Iran; soltanian@umsha.ac.ir; 3Department of Biostatistics and Epidemiology, School of Public Health, Hamadan University of Medical Sciences, Hamadan 6516848741, Iran; feryadresmohammad@gmail.com; 4Psychiatric Clinics, Division of Substance Use Disorders, University of Basel, 4002 Basel, Switzerland; Kenneth.Duersteler@upk.ch; 5Center for Addictive Disorders, Department of Psychiatry, Psychotherapy and Psychosomatics, Psychiatric Hospital, University of Zurich, 8001 Zurich, Switzerland; 6Center for Affective, Stress and Sleep Disorders (ZASS), Psychiatric University Hospital Basel, 4002 Basel, Switzerland; annette.bruehl@upk.ch (A.B.B.); dena.sadeghibahmani@upk.ch (D.S.-B.); 7Sleep Disorders Research Center, Kermanshah University of Medical Sciences, Kermanshah 67146, Iran; 8Department of Physical Therapy, University of Alabama at Birmingham, Birmingham, AL 35294, USA; 9Substance Abuse Prevention Research Center, Health Institute, Kermanshah University of Medical Sciences, Kermanshah 67146, Iran; 10Department of Psychology, Stanford University, Stanford, CA 94305, USA; 11Department of Sport, Exercise and Health, Division of Sport Science and Psychosocial Health, University of Basel, 4052 Basel, Switzerland; 12School of Medicine, Tehran University of Medical Sciences, Tehran 25529, Iran

**Keywords:** early ejaculation, male sexual dysfunction, lisdexamfetamine dimesylate, double-blind clinical trial, placebo

## Abstract

Background: Among male sexual dysfunctions, erectile dysfunction and early ejaculation have the highest prevalence rates. Here, we tested the influence of lisdexamfetamine dimesylate (Vyas^®^) on early ejaculation. To this end, we performed a double-blind randomized clinical trial among males with early ejaculation. Methods: A total of 46 males with early ejaculation (mean age: 35.23 years) and in stable marital relationships with regular weekly penile–vaginal intercourse were randomly assigned either to the lisdexamfetamine dimesylate condition (30 mg) or to the placebo condition. Compounds were taken about six hours before intended penile–vaginal intercourse. At baseline and four weeks later at the end of the study, participants completed a series of self-rating questionnaires covering early ejaculation. Female partners also rated participants’ early ejaculation profile. Results: Compared to the placebo condition, dimensions of early ejaculation improved over time in the lisdexamfetamine condition, though improvements were also observed in the placebo condition. Conclusions: Among male adults in stable marital relationships with regular weekly penile–vaginal intercourse, lisdexamfetamine dimesylate improved dimensions of early ejaculation. Given that improvements were also observed in the placebo condition, psychological factors such as increased attention to early ejaculation and favorable expectations of the compound should be considered.

## 1. Introduction

Among human beings as compared to all other species, penile–vaginal intercourse is outstanding in that it does not merely serve for reproduction [1,2,3]. As such, penile–vaginal intercourse among heterosexual partners serves a strong pair-bonding function [1,2,3]. Relatedly and unsurprisingly, among adults in heterosexual relationships, sexual satisfaction is closely linked to overall satisfaction with life [4,5,6,7] and couple satisfaction [8,9,10,11,12].

Among adults, sexual dysfunctions occur among both males and females, either as a single health issue or in combination with further somatic or psychological complaints [13]. Importantly, the ICD-11 shifted the emphasis from sexual dysfunction to “conditions related to sexual health”. The DSM-5 [14] defines a “sexual disorder” as sexual behaviors and experiences characterized as insufficient in quality, duration, or frequency. The definition is further qualified based on frequency of occurrence, namely 75–100% of sexual behaviors and experiences [15], which must be present for a minimum of six months, and which must cause significant distress (for a comprehensive overview, see [16]). If the duration is less than six months or it has not caused distress, it is diagnosed as a “dysfunction” [14,17]. Among women, sexual disorders include sexual interest/arousal disorder, genito-pelvic pain/penetration disorder, and female orgasmic disorder. Among men, sexual disorders include erectile disorder, male hypoactive sexual desire disorder, delayed ejaculation [18], and early ejaculation.

In the present study, we focused on male sexual dysfunction, and on early ejaculation among men with heterosexual orientation more specifically, given that compared to men with heterosexual orientation, homosexual men had 28% lower odds of reporting early ejaculation [19]. Typically, men reporting early ejaculation are unable to monitor their sexual arousal and sexual reaction. As a result, ejaculation occurs close to penetration of the vagina, or during or shortly after penetration [20]; thus, ejaculation occurs much earlier compared to the period of time related to sexual stimulation, sexual play and intercourse, and consensual accomplishing of sexual activity. Intravaginal ejaculation latency is often below one minute. Importantly, not always do men experience early ejaculation as a “disorder” or “dysfunction”; most often their female partners complain about losing their sexual interest under such conditions [20]. The prevalence rates of early ejaculation range from 2% to 30% [21], or from 5% [22] to 21% [23]. Further, according to Lotti and Maggi [24], as well as erectile dysfunction, male infertility appeared to be directly related to early ejaculation.

The etiological basis of early ejaculation remains elusive; both genetic and behaviorally acquired causes are discussed, and antecedent or reactive anxiety is often reported [20].

To treat early ejaculation, non-pharmacological and pharmacological treatments are employed. Non-pharmacological treatments consist of shifting the cognitive attention from sexual content to non-sexual content, although, according to Berner and Cockott [13], such interventions appeared to be anecdotal and accompanied by higher anxiety. Further techniques include squeeze and stop-start [25], sensate focus, stimulation devices, pelvic floor rehabilitation [26], and couples counseling [20,25,27]. Cooper et al. [26] concluded that evidence of the efficacy of physical behavioral techniques appeared to be limited and that behavioral therapies combined with drug treatments appeared to provide more encouraging results. In contrast, Fahrner and Kockott [27] reported that techniques such as squeeze and stop-start were effective.

Pharmacological treatments of early ejaculation include the use of serotonin reuptake inhibitors (SSRIs such as paroxetine, citalopram, escitalopram, sertraline, to name but a few), serotonin–noradrenalin reuptake inhibitors (SNRIs such as duloxetine), phosphodiesterase-5 (PDE5) inhibitors such as vardenafil and tadalafil, opioid analgesics, topical anesthetics–eutectics mixtures, and topical eutectics mixtures for early ejaculation [26]. In the present study, we focused on the efficacy of a pharmacological compound, specifically lisdexamfetamine dimesylate.

In the present study we focused on the influence of lisdexamfetamine dimesylate on early ejaculation for the following two reasons. First, Lyons et al. [28] reported in their case study that a 22 year-old Caucasian male with a history of pediatric pelvic neuroblastoma suffered from anejaculation and aspermia with normal sensation of climax. When the patient took lisdexamfetamine dimesylate 60 mg two hours prior to masturbation, he produced an antegrade ejaculate with the first use of the medication. Second, while the exact neurophysiological mechanism of neurotransmitter control over ejaculation is not fully understood, it appears that the dopaminergic pathway is associated with ejaculatory function [28,29,30]. Given this, we hypothesized that the intake of lisdexamfetamine dimesylate six hours prior to penile–vaginal intercourse might have a favorable effect on early ejaculation, compared to placebo. Lisdexamfetamine is prescribed to children, adolescents [31,32,33], and adults with attention deficit hyperactivity disorder (ADHD) [34,35], and to individuals with (binge) eating disorders [36,37,38]. We note that lisdexamfetamine dimesylate is a prodrug impacting on the dopamine transporter gene and thus impacting on the dopaminergic system: while lisdexamfetamine does not have a direct physiological impact, as an inactive prodrug it is converted in the body to dextroamphetamine, a pharmacologically active compound that is responsible for the drug’s activity (https://go.drugbank.com/drugs/DB01255; accessed on 12 May 2021).

Given this background, we assumed that compared to placebo, lisdexamfetamine would improve early ejaculation in men reporting this issue.

## 2. Methods

### 2.1. Study Design

Males with self-reported early ejaculation and treated at the outpatient clinic of Farshchian Hospital (Hamadan, Iran) were approached to participate in a randomized and double-blind clinical trial on the effect of lisdexamfetamine dimesylate (Vyas^®^) on early ejaculation. Participants were fully informed about the aims of the study, the study procedure and the confidential, secure, and anonymized data handling. All participants signed written informed consent forms. Participants were randomly assigned either to the lisdexamfetamine dimesylate condition or to the placebo condition. At the beginning and four weeks later at the end of the study participants completed a series of self-rating questionnaires covering sexual activity and ejaculation patterns. Female partners also rated participants’ early ejaculation profile both at the beginning and at the end of the study. The study was registered at the Iranian Register for Clinical Trials (IRCT registration number: IRCT20160523028008N10). The local ethics committee approved the study (registration number: IR.UMSHA.REC.1399.870), which was performed in accordance with the current and seventh revision [39] of the Declaration of Helsinki.

### 2.2. Participants

A total of 46 males with self-reported early ejaculation were assessed. Inclusion criteria were: 1. Male gender; 2: Age between 18 and 65 years; 3. Self-reported early ejaculation (see below); 4. Clinical diagnosis of early ejaculation based on early ejaculation diagnosis test (PEP; see below); 5. Stable heterosexual relationship; 6. Regular penile-vaginal intercourse at least twice a week for four consecutive weeks; 7. Compliance with study conditions; specifically: intake of the capsule about six hours before intended penile–vaginal intercourse; 8. Signed written informed consent. Exclusion criteria were: 1. Use of additional pharmacological and non-pharmacological techniques to cope with early ejaculation; 2. Use of condoms; 3. Neurological disorders such as multiple sclerosis (MS), neuromyelitis optica spectrum disorder (NMOSD) or Parkinson’s disease (PD), as ascertained by a thorough clinical-neurological interview; 4. Psychiatric disorders such as major depressive disorder, bipolar disorder, or substance use disorders, as ascertained by a thorough clinical psychiatric interview [40] based on the DSM-5 [14]; 5. Intake of psychoactive drugs such as antidepressants, mood stabilizers, sleep-inducing medications, or further medications impacting the dopaminergic system in an agonistic or antagonistic manner.

Of the 60 assessed, 46 (76.67%) were included in the study, and 40 (66.67%) completed the study. Statistics was performed per protocol. Figure 1 shows the flow chart.

### 2.3. Sample Size Calculations

We are unaware of a previous randomized and double-blind clinical trial with lisdexamfetamine for early ejaculation; given this, no a priori effect size could be used. However, we expected at least a medium effect size (partial eta-squared); the sample size calculation with G*Power [41] was as follows: partial eta-squared: 0.07; effect size f: 0.274; alpha: 0.05; beta: 0.8; number of groups: 2; number of measurements: 2; total sample size: 34. However, to counterbalance possible dropouts, the sample size was set at 46 participants.

### 2.4. Randomization

As in other studies [42,43,44,45], randomization was accomplished with randomization.com to create a list to assign 46 participants randomly to one of the two study conditions. Thereafter, a psychologist not otherwise involved in the study managed the assignments.

### 2.5. Measurements

#### 2.5.1. General Information on the Measurements

When available, we used translated and psychometrically tested Persian versions of a questionnaire. If a Persian version was not available, we translated the questionnaire and rigorously followed the algorithms proposed elsewhere [46,47]: Briefly, two translators independently translated the items into Persian. Second, the two versions of the translated items were compared. In the case of complete linguistic and semantic overlap, the translation was retained. When linguistic and semantic overlaps were low, a third translator endeavored to find the best linguistic and semantic fit between divergent translations. Two independent translators then back-translated the Persian versions into English. In the case of high linguistic and semantic overlap between the original English items and the back-translated version, the Persian items were accepted as the final versions. In the case of linguistic and semantic differences, both the Persian and the translated English version were adapted until high linguistic and semantic overlap was achieved.

#### 2.5.2. Demographic Information

Participants reported their age (years), duration of marriage (years), highest educational degree (compulsory school; diploma; high school degree; higher educational level such as bachelor’s, master’s or doctorate), number of children, and socioeconomic status.

#### 2.5.3. Information Related to Early Ejaculation

At baseline and four weeks later, at the end of the study, participants completed the following self-rating questionnaires:

#### 2.5.4. Number of Penile-Vaginal Intercourses per Week

At the end of the study, participants reported the average frequency of penile–vaginal intercourse per week (see Table 1).

#### 2.5.5. Premature Ejaculation Profile (PEP)

To further assess early ejaculation, participants and their female partners completed the Premature Ejaculation Profile (PEP) [48,49,50]. Questionnaire consists of items covering the following four domains: Sense of control over ejaculation; personal distress related to ejaculation; interpersonal difficulty; satisfaction with sexual intercourse. Answers are given on five-point Likert scales ranging from 0 (=very poor/not at all/none) to 4 (=very good/extremely/severe). Interclass correlation coefficients ranged from 0.66 to 0.83.

#### 2.5.6. Index for Premature Ejaculation (IPE)

The Index for Premature Ejaculation (IPE) is a self-report questionnaire developed to address issues (sexual satisfaction, control and distress) associated with early ejaculation [51]. The questionnaire consists of 10 items, and each item has 5 possible response options, ranging from 5 (=almost always/always/extremely distressed) to 1 (=almost never/never/not at all distressed). Items are aggregated to the following categories: sexual satisfaction, control, and distress; higher scores reflected higher satisfaction, higher control and lower distress.

#### 2.5.7. Intravaginal Ejaculation Latency Time (IELT)

To assess the time lapse between vaginal penetration and the intravaginal ejaculation, participants used a stopwatch. Intravaginal ejaculation latency time was reported in seconds [52].

## 3. Compounds

### 3.1. Lisdexamfetamine Dimesylate

Participants received a total of 12 capsules of 30 mg lisdexamfetamine dimesylate (Vyas^®^; manufacturer: Tadbir Kalaye Jam; Tehran, Iran). Participants were instructed to take a capsule six hours prior to the likelihood of penile–vaginal intercourse.

### 3.2. Placebo

Participants in the placebo condition received a total of 12 capsules of placebo. Participants were instructed to take a capsule six hours prior to the likelihood of penile–vaginal intercourse.

Placebo capsules consisted of lactose powder, glycerin, methylparaben, and propylparaben. Capsules of lisdexamfetamine dimesylate and placebo were identical in shape, color, weight, and smell.

### 3.3. Statistical Analysis

Sociodemographic data were compared between participants in the verum and placebo condition using a series of X^2^-tests and *t*-tests.

A series of ANOVAs for repeated measures was performed with the following factors: Time (baseline; study end; follow-up), Group (verum vs. placebo), and the Time x Group interaction; dependent variables were: Dimensions of early ejaculation, as reported in the questionnaires.

The nominal significance level was set at alpha < 0.05. Effect sizes for F-statistics were reported as partial eta squared (η_p_^2^), with 0.019 ≤ trivial effect size [T]; 0.020 < η_p_^2^ < 0.059 = small effect size [S], 0.06 < η_p_^2^ < 0.139 = medium effect size, [M], and η_p_^2^ ≥ 0.14 = large effect size [L]. Based on Becker’s approach to comparing mean changes [53], Cohen’s d effect sizes were reported for the pre- and post-trial changes within the two groups, and between the two groups at the end of the study. Effect sizes for *t*-tests were reported as Cohen’s d with the following ranges: d = 0–0.19: trivial effect sizes; d = 0.20–0.49: small effect sizes; d = 0.50–0.79: medium effect sizes; d ≥ 0.80: large effect sizes. All calculations were performed with SPSS^®^ 25.0 (IBM Corporation, Armonk, NY, USA) for Apple Mac^®^.

## 4. Results

### 4.1. General Information

Table 1 provides the descriptive and inferential statistical overview of participants in the lisdexamfetamine and placebo condition.

Participants in the lisdexamfetamine and placebo conditions did not differ regarding age, duration of marriage, number of children, number of penile-vaginal intercourses per week, socioeconomic status, and highest educational level.

### 4.2. Early Ejaculation-Related Information

Table 2 provides the descriptive statistical indices of early ejaculation-related information, while Table 3 reports the inferential statistical indices.

### 4.3. Premature Ejaculation Profile (PEP)

The premature ejaculation profile, as self-rated, increased over time (large effect size), but more so in the lisdexamfetamine (large effect size) condition compared to the placebo condition (large effect size of interaction).

The premature ejaculation profile, as rated by participants’ wives, increased over time (large effect size), with no group differences or with group differences over time (interaction); effect sizes were always small.

### 4.4. Index of Premature Ejaculation (IPE)

Satisfaction, control, and distress improved over time (always large effect sizes); no group differences (always trivial to small effect sizes) and no interactions were observed (trivial or medium effect sizes).

### 4.5. Intravaginal Ejaculation Latency Time (IELT)

Intravaginal ejaculation latency time increased over time (large effect size), but more so in the lisdexamfetamine compared to the placebo condition (large effect size of interaction). Scores were higher in the lisdexamfetamine compared to the placebo condition (medium effect size).

### 4.6. Mean Changes between the Lisdexamfetamine and the Placebo Condition at the End of the Study, and within the Lisdexamfetamine and Placebo Condition from Baseline to the End of the Study

Table 4 provides the overview of effect size calculations between the lisdexamfetamine and placebo conditions at the end of the study, and within the lisdexamfetamine and placebo conditions from baseline to the end of the study.

At the end of the study and compared to the placebo condition, lisdexamfetamine improved early ejaculation, as assessed using the premature ejaculation profile (self-rating) and intravaginal ejaculation latency (always large effect sizes), premature ejaculation profile as rated by female partners, and index of premature ejaculation control category (medium effect sizes). Small effect sizes were observed for index of premature ejaculation satisfaction and distress categories.

Within the lisdexamfetamine condition, large effect sizes were observed for premature ejaculation profile (self-rating and female partner’s rating), index of premature ejaculation control category, and intravaginal ejaculation latency. Medium effect sizes were observed for index of premature ejaculation satisfaction and distress categories.

Within the placebo condition, large effect sizes were observed for index of premature ejaculation satisfaction category; medium effect sizes were observed for Index of premature ejaculation distress category; trivial and small effect sizes were observed for premature ejaculation profile (self-rating and female partner’s rating), index of premature ejaculation control category, and intravaginal ejaculation latency.

## 5. Discussion

The key findings of the present study were that among males with early ejaculation, compared to a placebo condition, lisdexamfetamine dimesylate (Vyas^®^) improved dimensions of early ejaculation over a time lapse of four weeks when it was taken six hours before penile–vaginal intercourse. However, improvements were also observed in the placebo condition, suggesting that further unassessed dimensions such as focused attention and the placebo effect as a proxy of positive expectations must be considered. The present results add to the current literature in five ways: 1. Lisdexamfetamine dimesylate favorably impacts ejaculation patterns among men with early ejaculation; 2. Such improvements were also perceived by their wives; 3. Early ejaculation appears to be treatable with pharmacological interventions; 4. Improvements were observed also in the placebo condition, thus suggesting that psychological factors, at least in part, appear to be involved in the nature of early ejaculation and its treatment.

Despite the sparse literature on the effect of lisdexamfetamine on early ejaculation, we expected that compared to placebo, dimensions of early ejaculation would improve after regular intake of lisdexamfetamine during four consecutive weeks, and data did partially confirm this assumption. As such, the present data confirmed what has been observed in a case study of a 22-year-old Caucasian male with a history of pediatric pelvic neuroblastoma who suffered from anejaculation and aspermia with normal sensation of climax [28]: 60 mg of lisdexamfetamine dimesylate two hours before masturbation led to an antegrade ejaculate.

The quality of the data does not allow a deeper understanding of the underlying neuroendocrinological and cognitive–emotional processes underlying why the administration of lisdexamfetamine improved early ejaculation. Speculatively, the following six hypotheses are advanced:

First, there is some evidence that early ejaculation is related to the dopamine pathway; that is to say, a dysregulation in terms of inhibition of available dopamine concentration in the regulatory neurophysiological system appears to result in early ejaculation [29]. Results from studies with both human and animal subjects showed that central dopaminergic neurotransmission also appeared to be involved in the regulation of ejaculation. In rats, the stimulation of dopamine auto-receptors led to early ejaculation [54]; it appeared that the inhibition of DA neurotransmission resulted in early ejaculation. In humans, among a larger sample of 1290 men (mean age: 26.9 years), compared to carriers of the 9R9R/9R10R genotype dopamine transporter gene (DAT1), carriers of the 10R10R genotype dopamine transporter gene (DAT1) had a lower threshold to ejaculate [29]. Given this, Santtila et al. [29] concluded that dopaminergic neurotransmission was involved in ejaculation. Further, ejaculation was associated with primary activation in the mesodiencephalic transition zone, including the ventral tegmental area, which is involved with a wide variety of rewarding behaviors [30]. This rewards system, along with its emotional components of pleasure, joy and satisfaction and along with reward-enhancing action at a behavioral level, is highly associated with the availability of dopamine concentrations in the nigrostriatal, tuberoinfundibular, mesolimbic and mesocortical pathways [55,56].

To conclude, these findings suggest that dopaminergic neurotransmission is involved in ejaculation, in general, and that imbalance in available dopamine concentration is involved in early ejaculation, in specific.

Second, as regards the dopaminergic pathway of early ejaculation, lisdexamfetamine is highly related to increased availability of dopamine concentrations in the mesolimbic and mesocortical dopamine pathways. Note that lisdexamfetamine does not have a direct physiological impact, as lisdexamfetamine is an inactive prodrug. This prodrug is converted in the body to l-lysine and dextroamphetamine, the pharmacologically active metabolite which is responsible for the drug’s activity (https://go.drugbank.com/drugs/DB01255; accessed on 12 May 2021). Conversion takes place by enzyme hydrolysis in the red blood cells [57]. In rats, lisdexamfetamine increased mesocorticolimbic dopamine efflux and striatal dopamine D_2_ receptor occupancy [58]. Lisdexamfetamine is prescribed to children, adolescents [31,32,33], and adults with attention deficit hyperactivity disorder (ADHD) [34,35], and to individuals with (binge) eating disorders [36,37,38]. In individuals with binge eating disorder, lisdexamfetamine favorably impacted their emotional network to regulate emotional states [59].

Given the properties of lisdexamfetamine on the dopaminergic pathways and given that early ejaculation is also related to dysregulation of the dopamine pathway, it appears plausible that lisdexamfetamine favorably impacted on early ejaculation.

The background of the second hypothesis is as follows: Lisdexamfetamine improves cognitive performance, specifically long-term episodic memory, attention and inhibitory control [60,61]. It appears that the indirect activation of both dopamine receptor D_1_ and adrenoceptor α_2_ in the prefrontal cortex leads to such cognition-enhancing effects [60]. As such, the cognition-enhancing effect of lisdexamfetamine appeared to be comparable to the cognition-enhancing effect of methylphenidate [62]. Likewise, lisdexamfetamine also enhanced cortical network efficiency, which again was associated with improvements in working memory [63]. Importantly, both methylphenidate and lisdexamfetamine improved task saliency, that is to say, the motivation to perform a task, which in turn was associated with higher goal-directed behavior [64].

Given this background, the second hypothesis is: Lisdexamfetamine improved dimensions of early ejaculation because it improved cognitive control over the process of sexual activity, in general, and the timepoint of ejaculation, in specific. Gaining higher control over the timepoint of ejaculation coincides with cognitive–behavioral interventions upon early ejaculation: while men with early ejaculation experience ejaculation as a process beyond their behavioral and cognitive control, interventions such as the squeeze technique and the stop-start method confer to the individual (and his partner) higher control over the process of ejaculation [25,27]. Such techniques and interventions reduce helplessness and increase the feeling of control and self-efficacy [65]. As such, it is conceivable that the administration of lisdexamfetamine improved dimensions of early ejaculation via its cognition-enhancing effect in general, and its enhancing effects on task saliency, sustained attention, inhibitory control [60,61], and goal-directed behavior [64], in specific. However, if solely true, this hypothesis cannot explain the improvements in the placebo condition.

The basis of the third hypothesis is the concept of the Health Belief Model (HBM) [66,67,68]. Briefly, the Health Belief Model seeks to explain cognitive–emotional processes in terms of beliefs which underlie health-related behavior. As such, if sexual dysfunctions in general and early ejaculation in specific are considered “pure” physiological issues, then the administration of a medication should alleviate this sexual issue. In this case, unlike psychotherapeutic interventions, which presume a “psychological and psychosocial” basis for early ejaculation, the administration of a medical compound may coincide with a Health Belief Model (HBM) that early ejaculation is “merely” due to “biological” issues. The beauty of this hypothesis is that it also may explain the improvements in the placebo condition. In contrast, if it is solely true, one would not expect differences in early ejaculation performances between the lisdexamfetamine and placebo conditions.

The basis of the fourth hypothesis is complementary to the third hypothesis, and derives from placebo research: The beneficial effect of a compound does not derive from its chemical and neurophysiological structures per se, but from the cognitive–emotional expectations projected onto the compound [69,70,71]. As such, the “mere” expectation that early ejaculation could be “solved” might have triggered relief, decreased psychophysiological tension, and a favorable outcome. Indeed, improvements in early ejaculation were also observed in the placebo condition (see Table 2, Table 3 and Table 4). However, if solely true, this hypothesis does not explain why effect sizes were somewhat larger in the lisdexamfetamine condition (see Table 4).

Relatedly, fifth, only participants able and willing to comply with the study conditions were included in the study. As such, it is conceivable that the mere participation in such a study triggered positive expectations and improvements on early ejaculation. Research on psychotherapy showed that treatment motivation was a reliable predictor of improved psychotherapy outcome [72,73,74,75].

The basis of the sixth hypothesis is related to methodological issues: Latent and unassessed neuroendocrinological and cognitive–emotional factors could be responsible for the present pattern of results. For instance, it is conceivable that compared to the lisdexamfetamine condition, more participants in the placebo condition suffered from varicocele, the abnormal enlargement of the pampiniform venous plexus in the scrotum. There is some evidence to suggest that varicocelectomy might improve early ejaculation [76]. Thus, while highly unlikely, such methodological bias cannot fully be ruled out.

The novelty of the results should be balanced against the limitations mentioned above: First, while the overall pattern of results suggested *that* compared to placebo, lisdexamfetamine improved dimensions of early ejaculation, the quality of the data does not allow a conclusive answer as to *why* this happened. A further passive control condition and a wait-list condition might have helped to sort out possible placebo effects. Second, psychotherapeutic interventions recommend embedding interventions to treat early ejaculation within broader psychosocial and couples-related sexual therapies [13,25,27]. Third, with lisdexamfetamine early ejaculation latency time improved from about 33 s to about 60 s; mathematically, the latency time doubled, but from the point of view of a broader quality of sexual activity consisting of foreplay, mutual satisfaction and coordinated joint orgasm, 60 s might still be unsatisfactory. Fourth, it appears that early ejaculation is associated with a higher degree of anxiety and sociophobia [13,25,27]. As such, future studies should also assess such psychological dimensions.

## 6. Conclusions

Compared to a placebo condition, lisdexamfetamine dimesylate (Vyas^®^) improved some dimensions of early ejaculation in males with early ejaculation, though increased attention and placebo effects should be considered as well, as improvements were also observed in the placebo condition.

## Figures and Tables

**Figure 1 healthcare-09-00859-f001:**
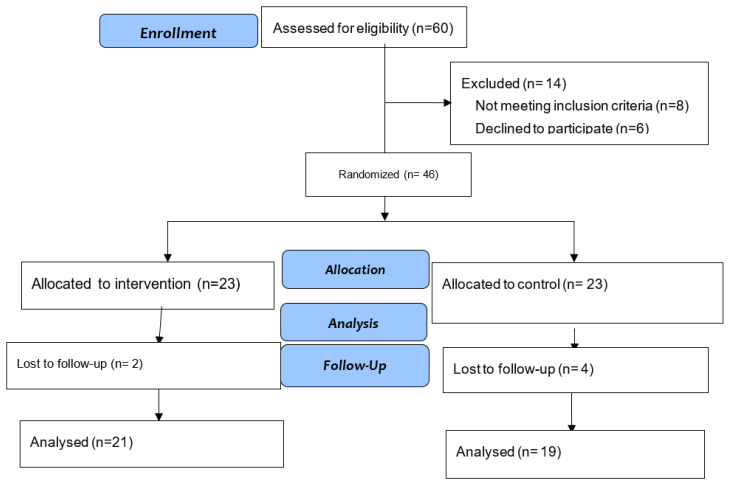
Diagram of the study design.

**Table 1 healthcare-09-00859-t001:** Descriptive and inferential statistical overview of sociodemographic baseline characteristics between participants in the lisdexamfetamine and the placebo condition.

Variables	Groups		Statistics
	Lisdexamfetamine	Placebo	
N	21	19	
	M (SD)	M (SD)	
Age (years)	34.66 (7.20)	35.84 (8.61)	t(38) = 0.64, *p* = 0.64
Duration of marriage (years)	9.04 (7.14)	11.10 (9.02)	t(38) = 0.43, *p* = 0.43
BMI (kg/m^2^)	26.99 (2.48)	26.51 (2.86)	t(38) = 0.57, *p* = 0.57
Frequency of penile–vaginal intercourse/week	2.35 (0.56)	2.45 (0.34)	t(38) = 0.24, *p* = 0.86
	*n*/*n*/*n*	*n*/*n*/*n*	
Children (0, 1, 2 or more)	12/2/7	9/2/8	X^2^(N = 40, df = 2) = 0.47, *p* = 0.98
Education status (no diploma, diploma, academic degree)	9/3/0	8/4/7	X^2^(N = 40, df = 4) = 0.35, *p* = 0.84
Socioeconomic status (poor, moderate, good)	5/8/8	5/11/3	X^2^(N = 40, df = 2) = 2.65, *p* = 0.27

**Table 2 healthcare-09-00859-t002:** Descriptive statistical indices of dimensions of early ejaculation, at baseline and at the end of the study 4 weeks later, separately for participants in the lisdexamfetamine and the placebo condition.

			Time Points	
N	Baseline		Study End	
	Lisdexamfetamine	Placebo	Lisdexamfetamine	Placebo
	21	19	21	19
	M (SD)	M (SD)	M (SD)	M (SD)
Premature ejaculation profile (self-rating)	3.38 (2.37)	2.73 (1.55)	6.57 (2.92)	3.72 (2.76)
Premature ejaculation profile (wives’ ratings)	4.23 (2.30)	3.8 (2.46)	6.47 (2.67)	4.89 (2.82)
Index of premature ejaculation; satisfaction	25.89 (15.84)	23.00 (10.03)	40.47 (22.15)	35.52 (23.01)
Index of premature ejaculation; control	22.61 (21.96)	20.72 (12.50)	45.21(28.60)	30.92 (27.12)
Index of premature ejaculation; distress	28.57 (25.04)	28.94 (27.65)	42.87 (21.85)	50.00 (30.51)
Intravaginal ejaculatory latency time (s)	32.85 (14.71)	31.57 (14.53)	55.71 (23.30)	34.47 (22.04)

Note: M: mean, SD: standard deviation.

**Table 3 healthcare-09-00859-t003:** Inferential statistical indices of dimensions of early ejaculation, at baseline and at the end of the study 4 weeks later, separately for participants in the lisdexamfetamine and the placebo condition.

	Factors					
	Time		Group		Time × Group Interaction	
	F	η_p_^2^	F	η_p_^2^	F	η_p_^2^
Premature ejaculation profile (self-rating)	30.08 ***	0.442 [L]	6.46 **	0.145 [L]	8.22 ***	0.178 [L]
Premature ejaculation profile (wives’ ratings)	11.49 ***	0.232 [L]	2.13	0.053 [S]	1.57	0.040 [S]
Index of premature ejaculation; satisfaction	25.09 ***	0.398 [L]	0.57	0.015 [S]	0.15	0.004 [S]
Index of premature ejaculation; control	18.83 ***	0.331 [L]	1.59	0.040 [S]	2.69 **	0.066 [M]
Index of premature ejaculation; distress	17.7 ***	0.318 [L]	0.27	0.007 [S]	0.66	0.017 [S]
Intravaginal ejaculatory latency time (s)	19.80 ***	0.343 [L]	4.49 **	0.106 [M]	11.89 ***	0.238 [L]

Note: Degrees of freedom: always (1, 38); [S] = small effect size; [M] = medium effect size; [L] = large effect size, * *p* < 0.05. ** *p* < 0.01. *** *p* < 0.001.

**Table 4 healthcare-09-00859-t004:** Overview of effect sizes; between group comparison at the end of the study; within group comparison from baseline to the end of the study.

	Effect Size Comparisons		
	Between the Lisdexamfetamine and Placebo Condition at the End of the Study	Within the Lisdexamfetamine Condition from Baseline to the Study End	Within the Placebo Condition from Baseline to the Study End
Premature ejaculation profile (self-rating)	0.99 [L]	1.19 [L]	0.45 [M]
Premature ejaculation profile (wives’ ratings)	0.60 [M]	0.9 [L]	0.38 [S]
Index of premature ejaculation; satisfaction	0.22 [S]	0.74 [M]	1.08 [L]
Index of premature ejaculation; control	0.52 [M]	0.88 [L]	0.48 [S]
Index of premature ejaculation; distress	0.27 [S]	0.61 [M]	0.74 [M]
Intravaginal ejaculatory latency time (s)	0.94 [L]	1.17 [L]	0.15 [T]

Notes: [T] = trivial effect size; [S] = small effect size; [M] = medium effect size; [L] = large effect size.

## Data Availability

Data are made available upon request from acknowledged experts in the field and upon a clear justification.

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
