# Peer review of "Influence of Lisdexamfetamine Dimesylate on Early Ejaculation—Results from a Double-Blind Randomized Clinical Trial"

_healthcare, 2021, doi:10.3390/healthcare9070859_

Round 1

Reviewer 1 Report

Through this well-designed and well-argued study, the authors have added new information to the current knowledge on the subject.

I have recommended additional resources for the authors to consider. Also, I have pointed some spelling errors and typos.

Author Response

We thank Reviewer #1 for the positive and encouraging comments, which helped us to improve the quality of the manuscript. Please find the detailed point-by-point-response attached as a separate file. 

The main changes are in bold characters highlighted in yellow. Other
minor changes have been made to improve grammar, style and word
choice and to correct punctuation.

Again, we thank you, the Handling Editor, the Editorial Board and the
anonymous reviewers for all your kind efforts. Of course, we are ready
to do more work on the files, in the event that the revision and the
answers are not yet entirely satisfactory.

Reviewer 2 Report

Dear Authors,

I congratulate with you for this paper. The topic is of interest and the article is well written. 

The introduction is well written and full of details. The length is a little bit too long, it should be summarized to emphasize results and discussion sections. 

Most information reported in the results section is already reported in the tables. I suggest the authors to simplify this section highlighting the most relevant results and explaining the reader the significance of a specific statistical result, without just reporting data already present in the tables. 

The discussion is even too much detailed. The description of the possible mechanisms of action of the drug and the causative mechanisms of premature ejaculation are correct, but should be shortened. I would rather explain the results of the other treatments for premature ejaculation and compare them with the study drug. 

I don't think you can state that the present results show that "the risk of addiction is low and side-effects.. are not observed" as the safety profile was not an outcome of this study and safety results were not reported. 

Author Response

We thank Reviewer #2 for the positive and encouraging comments, which helped us to improve the quality of the manuscript. Please find the detailed point-by-point-response attached as a separate file. 

The main changes are in bold characters highlighted in yellow. Other
minor changes have been made to improve grammar, style and word
choice and to correct punctuation.

Again, we thank you, the Handling Editor, the Editorial Board and the
anonymous reviewers for all your kind efforts. Of course, we are ready
to do more work on the files, in the event that

Reviewer 3 Report

Interesing manuscript, and very valuable project

In general introduction is redundant

In the introduction I think you refer to much the fact of heterosexual relationships, however, homosexual relationships are very similar. I think you should refer to relationships instead of heterosexual relationships.

Introduction start speaking about female sexual disorder, although highly valuable, you are talking about premature ejaculation. Better to talk about the pathophysiology of that and explain the definition base on the AUA

Figure 1 do not look ok (unable to see the participants)

For the results, in the table instate of report statistics, report the p-value

For table 2 add the p-values

Author Response

We thank Reviewer #3 for the positive and encouraging comments, which helped us to improve the quality of the manuscript. Please find the detailed point-by-point-response attached as a separate file. 

The main changes are in bold characters highlighted in yellow. Other
minor changes have been made to improve grammar, style and word
choice and to correct punctuation.

Again, we thank you, the Handling Editor, the Editorial Board and the
anonymous reviewers for all your kind efforts. Of course, we are ready
to do more work on the files, in the event that the revision and the
answers are not yet entirely satisfactory.

Round 2

Reviewer 3 Report

Authors correctly address the recommendations, however, I strongly recommend to add the p values in the statistics section on table 1 (there is no such think as a p value of 2.5, that is the X2 report -I assume-)

Author Response

Again, we thank Reviewer #3 for the scrutiny. As requested, we have added all p-values in Table 1. 

The main changes are in bold characters highlighted in yellow. Other
minor changes have been made to improve grammar, style and word
choice and to correct punctuation.

Again, thank you very much for the care devoted to thoroughly review the revised manuscript. 

This manuscript is a resubmission of an earlier submission. The following is a list of the peer review reports and author responses from that submission.